# Worldwide Detection of Informal Settlements via Topological Analysis of Crowdsourced Digital Maps

**Satej Soman** [1,*] 🔘 **, Anni Beukes** [1] 🔘 **, Cooper Nederhood** [1] 🔘 **and Nicholas Marchio** [1] 🔘
**and Luís M. A. Bettencourt** [1,2,3] 🔘

1   Mansueto Institute for Urban Innovation, University of Chicago, Chicago, IL 60637, USA;
    beukes@uchicago.edu (A.B.); cnederhood@uchicago.edu (C.N.); nmarchio@uchicago.edu (N.M.);
    bettencourt@uchicago.edu (L.M.A.B.)
2   Department of Ecology and Evolution, University of Chicago, Chicago, IL 60637, USA
3   Department of Sociology, University of Chicago, Chicago, IL 60637, USA
*   Correspondence: satej@uchicago.edu

**Abstract:** The recent growth of high-resolution spatial data, especially in developing urban environments, is enabling new approaches to civic activism, urban planning and the provision of services necessary for sustainable development. A special area of great potential and urgent need deals with urban expansion through informal settlements (slums). These neighborhoods are too often characterized by a lack of connections, both physical and socioeconomic, with detrimental effects to residents and their cities. Here, we show how a scalable computational approach based on the topological properties of digital maps can identify local infrastructural deficits and propose context-appropriate minimal solutions. We analyze 1 terabyte of OpenStreetMap (OSM) crowdsourced data to create worldwide indices of street block accessibility and local cadastral maps and propose infrastructure extensions with a focus on 120 Low and Middle Income Countries (LMICs) in the Global South. We illustrate how the lack of physical accessibility can be identified in detail, how the complexity and costs of solutions can be assessed and how detailed spatial proposals are generated. We discuss how these diagnostics and solutions provide a multiscalar set of new capabilities—from individual neighborhoods to global regions—that can coordinate local community knowledge with political agency, technical capability, and further research.

**Keywords:** OpenStreetMap; cities; slums; network analysis; remote sensing; human development; urban planning; GIS; cloud computing

## 1. Introduction

### 1.1. Context

The global population continues to urbanize quickly with nearly four billion people presently living in urban centers [1]. Of those, over 1.6 billion people live in under-serviced neighborhoods or informal settlements, commonly known as slums [2]. The number of people living in slums is expected to more than double in the next three decades if no action is taken [3]. The global distribution of this growth is far from uniform. Of the areas of the world delineated by the United Nations' Sustainable Development Goal Regions, Sub-Saharan Africa is unique in that its urban population is still expected to rise dramatically throughout the 21st century [4,5].

Approximately 294 million people lived in sub-Saharan African (SSA) cities and towns in 2010 [5,6]. In the next decade the sub-regions' urban population will grow to a projected 621 Million [6]. While mega-cities such as Lagos, Kinshasa, and Greater Johannesburg will continue to grow, it is expected that the most substantial proportion of the urban population do and will live in small and intermediate size urban centres [6–8], where local governments lack responsive structures, suffer inadequacies in the provision of basic services and have limited capacity to fulfill their responsibilities to provide good living conditions and good health for their residents [6,9,10].

Ensuring sustainable and equitable development in most of these cities will require massive collective and coordinated action across local to global scales. The role that residents and participatory practices need to play in this context does not only depend on better and more accessible data, but also on our ability to turn these data into actionable intelligence to support decision making and planning at multiple scales [11,12].

Expanding quality-of-life and economic development opportunities for the inhabitants of informal settlements depends on an accurate picture at the city level of which neighborhoods are critically under-serviced and which could prosper immediately with small and focused upgrading interventions [13]. Though determinations of whether a neighborhood is under-serviced depends greatly on local concerns, a topological analysis of building footprints in relation to the formal road network offers a quantitative criterion [14]. With a topological approach, it is possible to determine the number of building parcels that must be crossed to access the existing street network (and therefore, emergency services, sanitation, clean water, etc.). Focusing on the topologically-invariant properties of maps also allows us to analyze cities without developing approaches that depend on detailed morphology or visual clues, which are all context- and history-dependent [14].

Here, we describe a large-scale computational workflow to detect informal settlements anywhere in the world. We describe methods to apply topological analyses to planet-scale datasets and provide illustrations based on OpenStreetMap data, including automatically generated scenarios for minimally disruptive infrastructural interventions that provide universal street access to all buildings within informal settlements.

*1.2. Prior Work and Novel Contributions*

There has been an unprecedented effort to exploit high-resolution remote sensed data, street view images, machine learning, and novel spatial data technologies to understand how cities are growing and to detect the location of informal settlements [15–17]. The most recent advances in data-driven techniques like deep learning have shown the superiority of learning from data over hand-designed features dependent on human experts [18]. Deep learning approaches like convolutional neural networks (CNNs), having achieved success in fundamental computer vision tasks, are being used in a variety of ways to understand the built environment. For example, researchers use satellite imagery and CNNs to build a poverty map of sub-Saharan Africa [19]. Others use satellite imagery and CNNs to compare socioeconomic outcomes across various states in India and use deep neural networks to analyze patterns in urban land use [20,21]. Deep learning approaches have also gained traction in the slum mapping task. For example, Ref. [22] use a unique time-series of imagery to perform change detection in slums in Mumbai. However, data-driven approaches, especially those utilizing deep learning, require huge amounts of labeled training data, which is especially difficult to obtain for typically data-scarce slums. Other efforts, like the 2018 Deep Globe Challenge, provide new public datasets to spur research in bringing deep learning to remote sensing but datasets for slum identification remain limited in applicability [23]. Further, validating neural network classification performance to unseen out-of-sample imagery from distinct geographies remains an open question [24]. Finally, while the features learned by a neural network,

given sufficient data, should ultimately offer superior prediction, they are often difficult to interpret and contingent on choices of labeled training sets [25].

Traditional approaches relying on hand-designed features, while theoretically inferior in prediction power, are more easily interpretable, which is critical when using research to drive real-world planning and policy decisions. Most of these approaches use a two-step strategy. First, satellite images are pre-processed to remove artifacts (such as accounting for sensor distortion and applying line segment- and edge-detection algorithms [15]) and extract key discriminative features. Across both deep learning-driven and hand-tuned analyses, image texture features are the most promising in generalizing beyond specific geographic areas of study [26]. Examples of texture features include pixel-scale features such as entropy, contrast, variance, and mean pixel values [27], while manually-labeled features include green ratio [16], relative area of neighboring objects [16], lacunarity [15,28], building height [17], and heterogeneity of building orientations [17]. Features may also be extracted at multiple spatial scales, since many identifying aspects of slums are scale-dependent [26]. Second, the resulting raster images and extracted features are compared to given definitions of informal settlements (slums) in specific urban areas to identify distinctions in feature space between slums and the broader urban fabric [26,29]. Specifically, because slums are often small, dense, and composed of non-regular patterns, they exhibit a higher fractal dimension than formal neighborhoods [28]. These automated approaches best identify large, dense slums in major urban areas, but may miss smaller or more peripheral communities which do not share the same geometric signature [29].

Because all of these approaches rely on human-labeled features they often generalize poorly to new contexts that differ from the original training data. Slums in different parts of the world and at different growth stages exhibit different geometric and topological patterns. Thus, adapting these methods to new contexts typically requires careful re-tuning of threshold values and redesigning of heuristics and logical statements. For example, early object-based approaches for slum identification in Cape Town, South Africa used rules-based on physical characteristics like building size, which then needed to be updated when adapting the approach to Rio de Janeiro [30]. Particular studies of the Indian cities of Pune [16] and Hyderabad [15] also note their inapplicability to other cities. The expert knowledge required to adapt these methodologies to a new context represents a significant challenge to scaling up these methods globally and to peri-urban contexts. A recent survey of remote sensing approaches to slum mapping concludes that methods to date "tend to be concentrated to a few geographical areas and often focus on the use of a single approach" [29].

In contrast to these approaches, we apply here a topologically-invariant measure of formal infrastructure access to open source vector data consisting of two elements: building footprints and street networks [31]. This approach defines informal settlements as (sets of) street blocks lacking physical access to formal infrastructure (proxied by the street network). The street network naturally divides the city into blocks, in which buildings are enclosed by accesses (streets) and other boundaries. The topological approach to the analysis of the accessibility of street blocks is concerned only with the relationships between street accesses and buildings, not detailed geometry. As is well known, these relationships translate into mathematical graphs with algebraic properties that can be computed in fast and efficient ways by a variety of existing algorithms.

We argue that this approach—based on the general mathematical properties of graphs—offers greater generalizability. We show how it can be applied anywhere in the world to not only classify informal settlements but, more importantly, to provide measures of the complexity of their access deficits (a slum "spectrum") and design specific minimally disruptive solutions.

For the data required to calculate these indices, we turn to OpenStreetMap, a crowdsourced, open-source geospatial database with vector information on buildings, amenities, services, infrastructure, street networks, public parks, and various other cartographic features [32]. Despite the fact that data in OpenStreetMap are generated collaboratively, there are reasons to believe the data are reasonably complete

and current. A survey of crowdsourced cartography efforts estimates that 80% of the world's road network is accurately reflected in open-source geospatial databases [33]. Additionally, civic activism to improve the cartographic resources of informal settlements has used OpenStreetMap as a ready-made database to formalize and collate grassroots mapping efforts [34]. In some cases, local organizations in countries like Lesotho rely on OSM as an authoritative repository of building footprint and tenure information [35]. Raster-based approaches that use existing maps of slum areas as ground truth for calibration may also run into issues identifying new slums lacking official designation. While large slums have largely disappeared from high income nations, we observe that rural-to-urban migration continues in developing countries [36] and these migrants may not settle in areas that match the textural characteristics of previously-identified slums. By focusing on infrastructure access, we can identify new areas in which building construction and land use outpace service and infrastructure delivery.

The topological and computer-vision approaches are not mutually exclusive. The methods described here can use the building footprints and road network geometries obtained through either texture-based or object-oriented analysis (OOA) of Earth observation data, paving the way for a hybrid approach in which computer vision techniques automate much of the annotation, while local knowledge is factored in via manual corrections of detected objects. A morphological study of informal settlements in 44 cities serves as an example of this approach: while much of the modelling was done algorithmically, local knowledge about building structure was added via geotagging and in situ surveys [17]. Recent studies in slum mapping in Kenya used image features in satellite data, georectified locations of critical services, online real estate data, and many others [37], showing the range of complementary data sources available in slum mapping efforts.

Once infrastructure deficits are identified, the next natural question is the development of solutions that are both cost-effective and respectful of local community priorities. "Reblocking" is a process of extending access networks to provide universal access to buildings within a street block. The topological structure of the buildings within a block provides a constrained-optimization formulation of this problem: the goal is to find a minimal path length on the graph that connects every parcel to the boundary of the street block. Notably, the solutions to this optimization problem mirror the in situ upgrades that residents historically have chosen to implement, and also provide a class of solutions to offer communities the ability to choose which infrastructure upgrade plan best suits local conditions and constraints [31].

Previous reblocking implementations built a statistical distribution of potential path components and used Monte Carlo sampling methods with an importance metric exponentially proportional to path segment length for the purpose of iteratively building a class of minimum-cost access networks [31]. In our investigation, we found that posing this problem in terms of Steiner tree generation provides similar access networks while being much more computationally feasible to implement at country- and planet-scale.

## 2. Materials and Methods

### 2.1. Geospatial Data and Computation

The data universe for our work involves three different entities: administrative boundary polygons, building footprint polygons, and street network line string collections. The administrative data boundaries were sourced from the Database of Global Administrative Areas (GADM). Building footprints and road networks were sourced from OpenStreetMap (OSM) data. Since OSM does not provide direct downloads, the data under consideration was obtained from a mirror hosted by GeoFabrik. The data used in the analysis and figures in this paper reflect a snapshot of OpenStreetMap as of July 2019, focusing on 120 countries that fall within the United Nations' definition of the Global South. The methods and analysis described are general and can be applied to any similar geospatial data at the block level, e.g., from historical sources or high income nations.

The input geospatial data were large, involving potentially every building footprint and street worldwide. However, this data could be naturally parallelized at the city block level. The use of GADM administrative boundaries provided a convenient system to subdivide large spatial files into smaller units more amenable to parallelization because of their standardization and typical unit size. To select the relevant features, we queried a number of metadata tags in the GeoFabrik extract (listed in Table 1). The resulting building footprint `Polygons` and street network `LineStrings` in our extract exceeded 1 terabyte in size, making serial computation impractical. To process the input data in parallel, we intersected the most granular administrative boundaries available for each country with the corresponding building footprint and street network geometries (see Figure 1). We then ran our block geometry extraction for each administrative unit, and then further parallelized our cadastral map generation and planar graph construction (all detailed in sections below) at the level of each street block. Using the University of Chicago's Midway2 distributed computing cluster (managed by the University's Research Computing Center), we processed 72,896 administrative units across 120 countries comprising the Global South, corresponding to 4,343,602 street blocks. The computation was parallelized across Intel E5-2680v4 2.4 GHz processors on a Scientific Linux 7.2 operating system using a SLURM resource scheduler. The node specifications consisted of 28 cores with 50 GB of allocatable RAM.

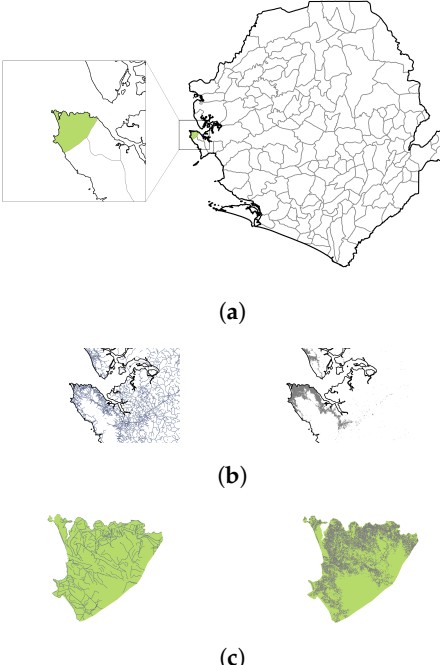

(**a**)

(**b**)

(**c**)

**Figure 1.** Parallelization scheme of geospatial data on streets and building footprints. (**a**) For each country, the most granular administrative boundaries are obtained from the Database of Global Administrative Areas (GADM); the country border of Sierra Leone is shown in black, while administrative boundaries are in grey. Highlighted in green and in inset is the administrative unit containing known informal settlements in the capital city of Freetown. (**b**) The national street networks layer (left, blue lines) and building footprints layer (right, gray polygons), in the region around Freetown, extracted from GeoFabrik data. Only the top 20% of roads by `LineString` length and top 20% of buildings by `Polygon` area are shown. (**c**) Each country-level layer is intersected with the boundaries of each administrative unit to obtain unit-level extracts of road networks (left) and building footprints (right). The block extraction and topological analysis, detailed below, is then run on the unit-level extracts in parallel.

**Table 1.** GeoFabrik ontology layers and associated metadata tags used in analysis.

| Layer | Tags |
|---|---|
| lines | natural = 'coastline' |
| | non-null waterway |
| | non-null building |
| multipolygons | non-null building |

### 2.2. Street Block Geometry Determination

To extract the geometry of each city block, we examined the street network within each administrative network (Figure 2a). The individual road segments were unioned together to create a single object representing the road network geometry (Figure 2b). The difference between the administrative unit geometry and the overall road network geometry was comprised of the polygons we wish to extract (Figure 2c). To isolate these polygons, we used a set-theoretic difference between the administrative unit's area and the road network geometry buffered by a small amount (to render it two-dimensionally). An alternative approach is to identify all road intersections and search for loops within a graph of nodes representing road intersections, but we found this implementation to be less computationally performant.

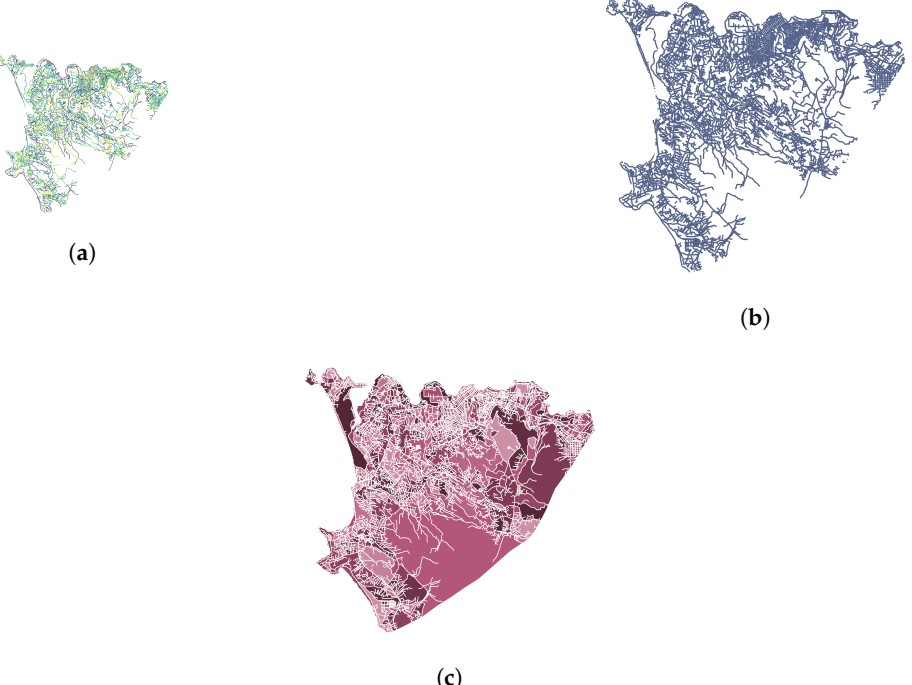

(**a**)

(**b**)

(**c**)

**Figure 2.** City block extraction. (**a**) The set of streets for an administrative unit, formatted originally as individual `LineStrings` (shown here in varying colors for distinction), comprising a regional street network. (**b**) The union of individual streets (which results in a `MultiLineString` object), buffered by a small amount to create a `MultiPolygon` (shown here with an exaggerated buffer). (**c**) The difference between administrative unit boundary and the road network geometry gives the geometric description of street blocks. Colors distinguish between adjacent street blocks. All subfigures show the street network and block geometries for Freetown, Sierra Leone.

### 2.3. Land Parcel Map Construction

Given the geometry of each street block, we created a corresponding land parcel cadastral map, as seen in Figure 3. To do this, in each street block every building footprint was assigned a subset of the block geometry, determining the precise dimensions and area of each parcel. This constituted a proposal for a possible land cadastre that could be checked and corrected against official records where available.

In initial topological analyses, actual cadastral maps were used. These were either obtained from local tax authorities, or via community- or NGO-generated mapping efforts [31]. To scale up the analysis to general OSM data, we developed a method for automatically generating parcels anywhere in the world. Note that the inclusion of parcel boundary records in crowdsourced maps such as OSM, is technically possible and is being considered, but is presently unavailable [38].

To create land parcel maps, we first intersected each street block polygon with building footprints to isolate the set of buildings which fall within the block boundaries. Then, in the absence of additional knowledge, we adopted a Voronoi decomposition of the street block geometry taking each building footprint as a node. The Voronoi algorithm provides the most egalitarian land assignment in the sense that all land in the block closest to each building (relative to other buildings) is assigned to it. A methodological issue arose from the fact that footprints were shapes, not points. Thus, a classical decomposition of street block polygons with building footprint centroids often resulted in cadastral boundaries that intersected the building shapes. To resolve this issue, we generated a Voronoi decomposition of the street block polygon using the building footprint vertices and regularly spaced samples of footprint boundaries as segmented input points. Each resulting Voronoi cell whose centroids were vertices of the same building were then unioned by unique building identifiers to create a land parcel assignment that did not intersect the building footprints and that ran in general at an equal distance between adjacent buildings.

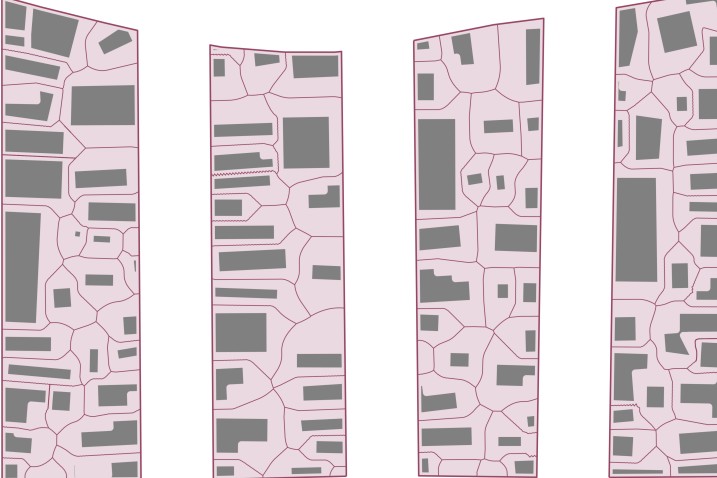

**Figure 3. Results of cadastral approximation process.** Each light crimson enclosure is a street block from Freetown, with building footprints overlaid in grey, and internal parcel boundaries shown in thinner crimson lines. This illustrative example was programmatically generated with a modified Voronoi decomposition.

### 2.4. Estimation of Street Block Topology and Access to Buildings

Next we analyzed and classified each street block in terms of its level of accessibility to each land parcel from the existing street network. We used the methods developed in [31], which express the

adjacency of parcels within a block to each other in terms of a graph. Once in this form, the operations that determine the level of accessibility to the parcels in each block become purely algebraic.

We describe this procedure here for completeness; see Figure 4 for the general idea, and Figure 5 for an application to street blocks in Freetown. Briefly, cadastral land parcels were taken as the initial faces of a planar graph, $S_0$. This planar graph structure had a "weak dual"—a corresponding graph formed from connecting the centroids of each face to each adjoining parcel's centroid. We term this structure a weak dual graph because incomplete faces were discarded, and so taking the weak dual of the weak dual did not restore the original graph. Instead, successively taking weak duals formed a sequence of planar graphs $(S_1, \ldots, S_k)$. Every sequence of planar graph weak duals simplified the graph and eventually converged to a trivial tree graph, without internal loops (faces). The number of weak dual iterations required to achieve this terminal trivial graph (or the length of the weak dual sequence), is an integer, $k$, which measures the spatial accessibility of parcels in that block [31]. The higher the number of weak dual iterations, the higher $k$ becomes, expressing that we are dealing with a more complex block with more layers of internal parcels. The least accessible parcel is $k/2$ layers (intervening) parcels and their buildings, from existing street accesses [14]. For this reason, street blocks with $k > 2$ contained inaccessible buildings and could be considered an informal settlement.

This approach had two advantages. First, it reduced the problem of dealing with varied morphologies (shapes) and building densities across street blocks to a simpler comparison of graphs and numbers expressing relationships between parcels and the street network. Second, this approach located precisely within a street block where the least accessible areas were by examining the parcels nearest the final tree graphs; see Figure 2c.

### 2.5. Map of Inaccessible and Under-Serviced Neighborhoods

The street block by street block classification of inaccessibility of parcels measured by the $k$-index was then visualized as a GIS layer in terms of polygons of different colors. This is shown for a section of Nairobi, Kenya in Figure 6. Such maps allow for the visual inspection of areas that are inaccessible by public accesses and thus, almost in every case, lacking public services such as water and sanitation, as well as emergency services.

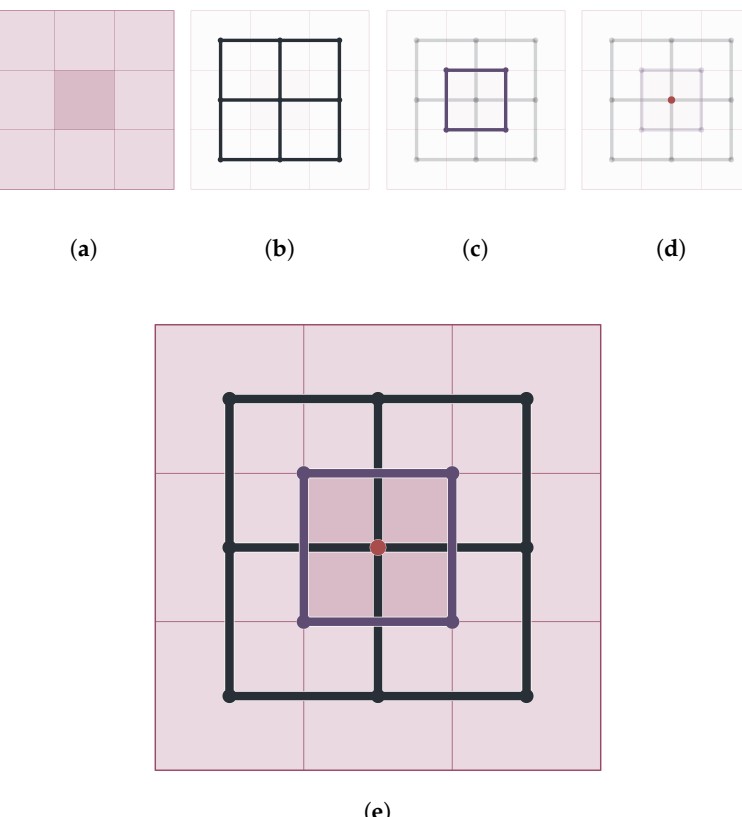

(e)

**Figure 4.** Successive weak duals of planar graphs. From each block's parcels, we can construct successive weak duals of a planar graph until the resulting structure forms a trivial network (a tree). The number of weak dual operations required to reach a tree is termed the *k*-index for the block. Two blocks with the same *k*-index have the same level of infrastructure access for their enclosed buildings, regardless of block shape. (**a**) Once the parcels are generated for a block, the parcel polygons are used as the faces of a planar graph ($S_0$). Note that the central parcel, shaded darker, has no access to the surrounding road network. (**b**) $S_1$ in black. (**c**) $S_2$ in purple. (**d**) $S_3$ in red, a trivial 1-node graph, indicating $k = 3$. (**e**) Superimposed weak dual sequence for the original parcels.

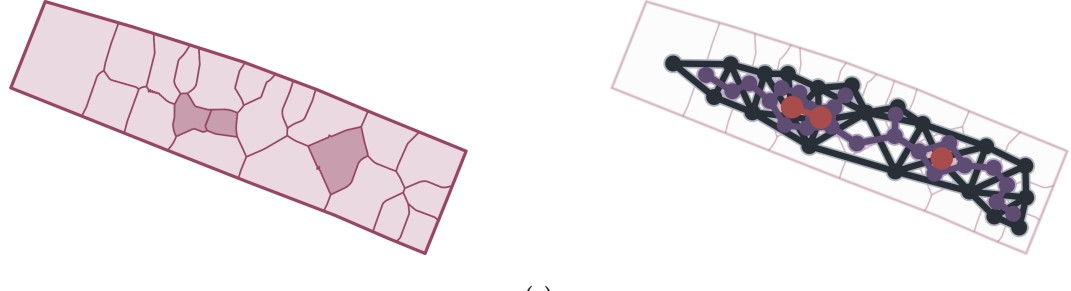

(a)

**Figure 5.** *Cont.*

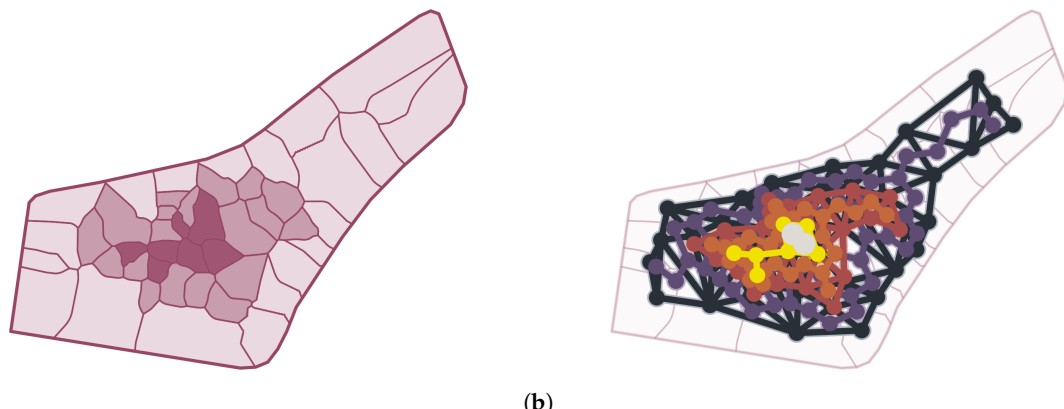

(**b**)

**Figure 5.** Weak dual graph sequences for Freetown street blocks (**a**) A street block in Freetown with inner parcels shaded (left), and its corresponding weak dual sequence (right; black → purple → red), of length 3 (*k* = 3). (**b**) A more complex, and thus less accessible, Freetown block with multiple levels of inner parcels shaded (left), and its corresponding weak dual sequence (right; black → purple → red → orange → yellow → white), of length 6 (*k* = 6). Due to the higher *k*-index, this block would be considered more "slum-like".

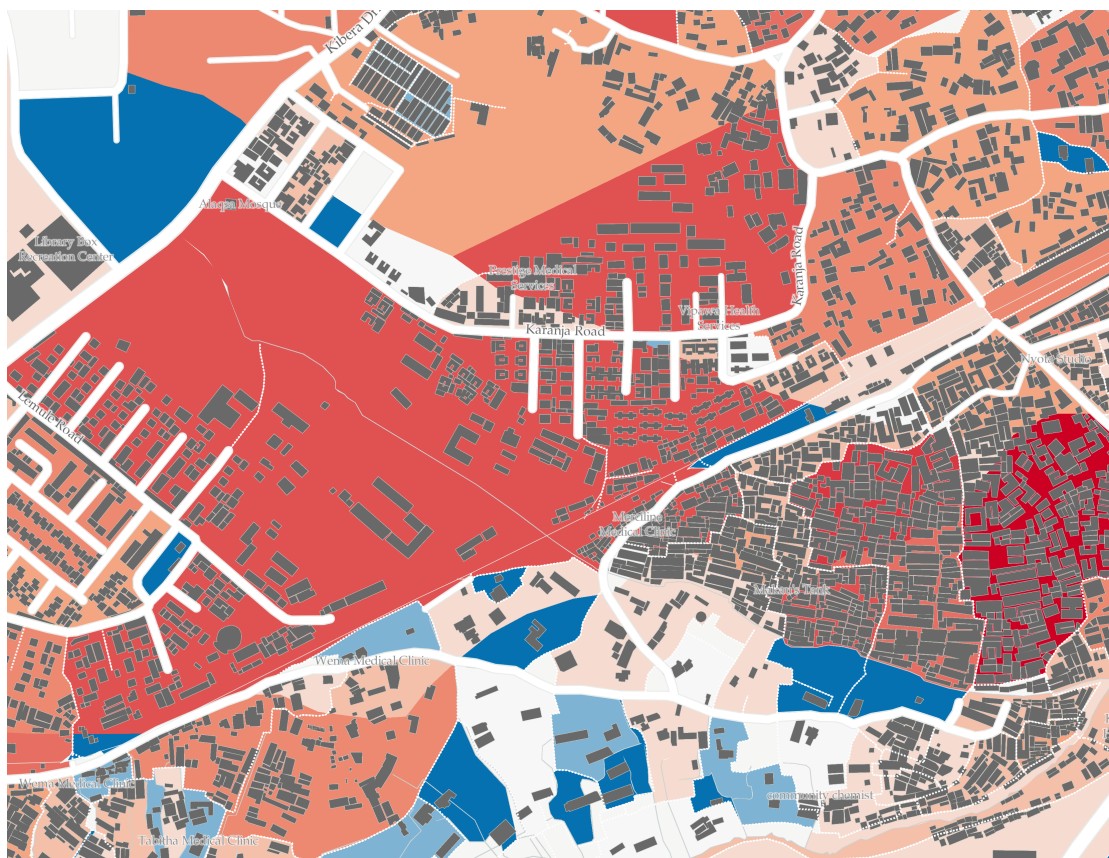

**Figure 6.** Choropleth of street blocks by complexity. Blocks in Kibera, a slum in southwest Nairobi, Kenya, colored according to block complexity. Blocks with relatively low complexity (*k* < 2) are colored blue as each of their enclosed buildings (in grey) have direct access to the surrounding road network. Progressively lighter shades of blue indicate worse access, while shades of orange and red indicate severely impacted access (red for *k* ≥ 12). This image was exported from the Mapbox Studio backend using OpenStreetMap data, ©Mapbox, ©OpenStreetMap. To view the full interactive map, visit MillionNeighborhoods.org.

*2.6. Estimation of Minimal Street Network Extensions for Universal Connectivity*

The *k*-complexity index provided a measure of public access to each parcel in every street block. We could therefore create a process of constrained optimization for the block's access network based on the value of *k*. For initially inaccessible street blocks, this process identified the minimal (in the simplest case, measured in terms of additional street length) extensions of existing accesses that rendered a block universally accessible. This minimal extension was unique, while other options involved improvements in the location and shape of street extensions at the cost of additional length.

To implement these formal concepts in practice, we started from the fact that cadastral maps included both existing streets and the parcel boundaries between buildings. We assumed that any parcel boundary was a candidate for a new street segment. Then, we could search for the minimal cost street network that made $k \leq 2$, by activating the minimal set of parcel boundaries that connected initially inaccessible parcels to the existing street network. This process is commonly called "reblocking" in the development literature and is a recommended method to develop infrastructure within informal settlements [31].

To implement minimal reblocking as an algorithm, we posed it as an optimal subgraph identification problem. Given an undirected graph $G = (V, E)$ with non-negative edge weights, *e*, and a subset $S \in V$ of target nodes, we seek the minimal-weight tree that spans the target nodes *S*. This is a type of Steiner tree problem that is known to be NP-hard [39]. Therefore, we approximated the solution by computing the minimum spanning tree of the metric closure of the subgraph induced by the target nodes *S*. The metric closure was the complete graph in which each edge weight is the shortest path distance between the corresponding nodes [39].

To pose reblocking as a Steiner tree problem we took the parcel boundaries as an undirected graph with edge weights equal to the Euclidean distance between nodes if connected. Each face of this graph contained a building footprint: we projected the centroid of each building footprint to the nearest edge of the graph as the assumed point of access (which could instead be set by real world preferences). The collection of projected building centroids served as our target nodes, *S*.

Before estimating the solution, we performed various modifications to the graph. We removed as targets those building centroids which already bordered an existing street, as those buildings already had full connectivity. We added a target node to the exterior street network such that the estimated network between all residences connected to the external street network. This simplified the graph and decreased the size of the Steiner tree problem. All edges corresponding to an existing street were given a weight of zero. A node with exactly two edges captured a bend in the parcel boundary but had no meaning in a graph context, and therefore we removed such nodes and combined their distance with that of its neighbor. We approximated the optimal Steiner tree by solving the minimum spanning tree of the metric closure of the subgraph induced by the target nodes, *S*. The result of this procedure is shown in Figure 7, where additional accesses are shown as white lines.

The entire procedure allowed us to start with any large digital map of building footprints and street networks, and automatically decompose it spatially in parallel units, determine accessibility of each street block and estimate minimal local infrastructure improvements that may allow an informal settlement to become universally accessible, like formal neighborhoods. Next we describe some results of applying this work stream to large urban maps.

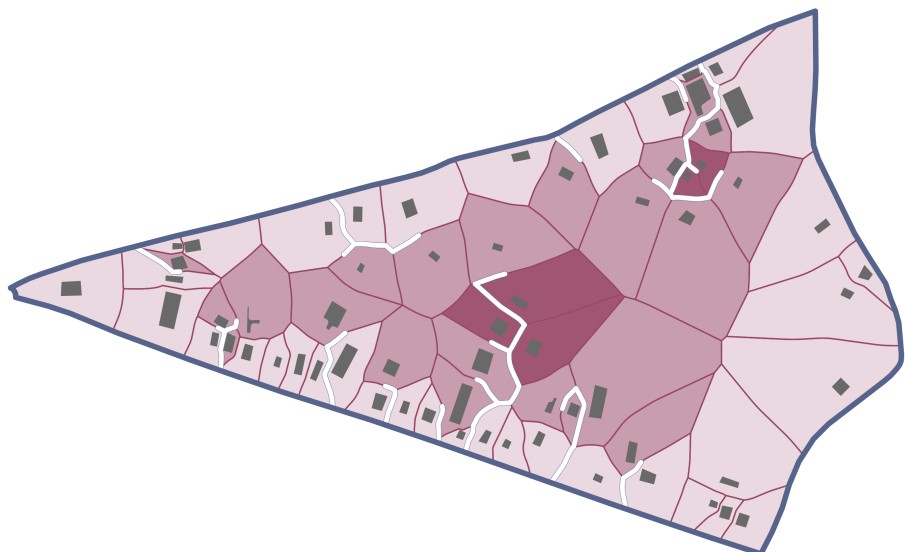

**Figure 7.** Reblocked block in Freetown, Sierra Leone. Using the graph structure generated from the cadastral parcels, we can determine how many parcels are not adjacent to the block boundary, and calculate the least-cost network of additional street segments (white) that will connect every building (gray) to the existing street network (blue). We pose this as a Steiner tree approximation problem and estimate the smallest length set of parcel edges that, if turned into streets, would connect every building to the boundary and reduce the block's $k$-index.

## 3. Results

The methods just described allowed the systematic analysis of service accessibility for street blocks anywhere in the world. For blocks where many land parcels are inaccessible, and given other contextual variables, we could assert with a large degree of confidence that we were dealing with an informal settlement. We performed these checks in many parts of the world where informal settlements were identified by local authorities, NGOs or resident communities, but also note that the methods often found additional cases, especially in peri-urban areas as discussed below.

The distinct strength of the methods described here is not so much the binary classification of whether a neighborhood is an informal settlement (or slum), but the detailed characterization of the difficulty of creating local accesses and associated costs, as well as the creation of basic detailed spatial proposals to address these deficits.

Then, by aggregating all the street blocks that belong to a city or administrative region (municipality, district, nation) we can provide a spatially resolved analysis of needs and scope of potential solutions across scales. Different spatial scales are in practice associated with local information, political agency and aggregate infrastructure funding, so that a multi-scalar analysis has the potential to be transformative.

We now provide a few illustrations of this work using crowdsourced data from OSM.

### 3.1. Analysis of Complexity and Other Block-Level Metrics

With blocks extracted, land parcels defined, and reblocking plans created, we can explore the empirical relationships between the block complexity index $k$ and other metrics. In particular, this analysis allows us to see that any attempt at a binary classification of neighborhoods as slums or not slums is too crude to be useful in practice. Instead, we should appreciate that there is a broad spectrum of lack of accesses and informality. To his end, we focused on two urban areas in West Africa, with especially complete OSM data collections: Freetown, the capital of Sierra Leone, and Monrovia, the capital of Liberia. These two

cities have been the target of various crowdsourced mapping efforts associated with Humanitarian OSM, especially during the Ebola outbreaks of 2014–2016.

Figures 8b and 9b show that the majority of street blocks lacking accesses in both cities were relatively simple, characterized by $k \leq 5$–6 and by a few hundred buildings each. This type of street block constituted the bulk of reblocking challenges in terms of land area and population in both cities and in other cases we analyzed. These Figures also show how the two cities were somewhat different, with Freetown manifesting a somewhat more challenging picture in terms of block complexity distribution.

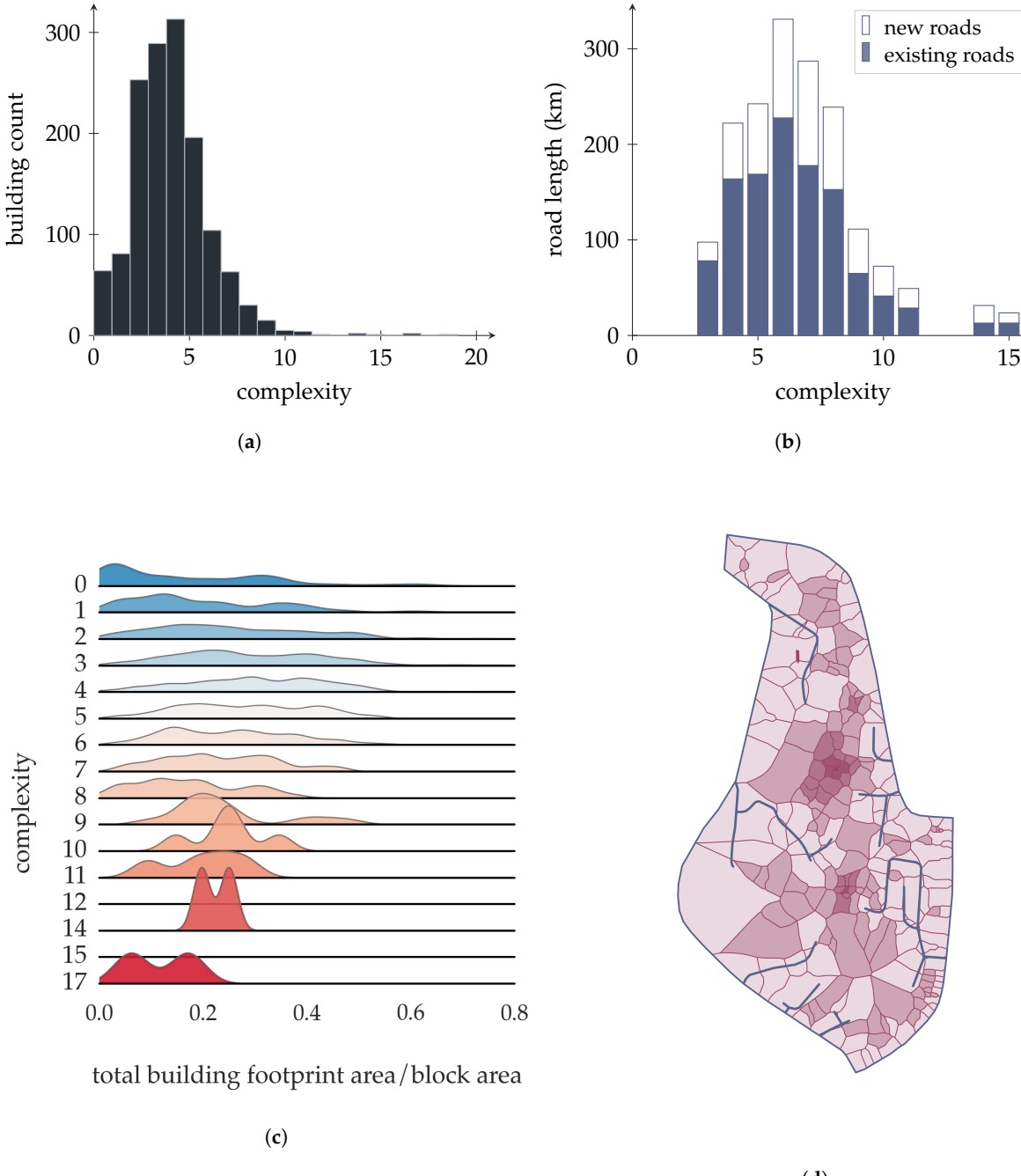

**Figure 8.** Analysis of block characteristics for Freetown, Sierra Leone (**a**) Histogram of buildings by block complexity. (**b**) Distribution of proposed new road construction by block complexity. (**c**) Kernel density estimates for distributions of building-block areal ratios at each value of *k* observed (denoted "complexity" in the figure). (**d**) Map of Freetown administrative campus containing Parliament and other civic buildings in which limited street access occurs by design.

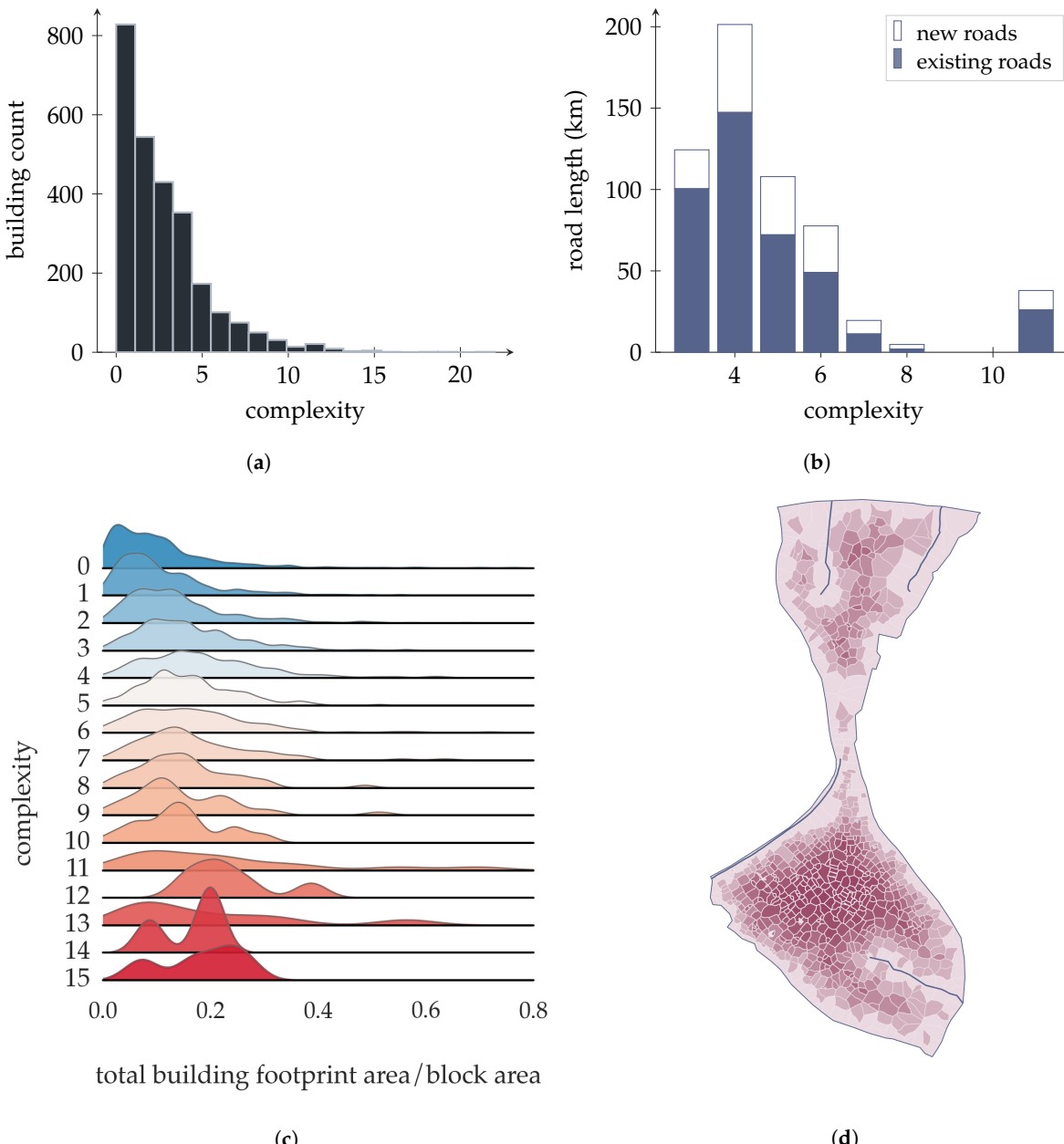

**Figure 9.** Analysis of Block Characteristics for Monrovia, Liberia. (**a**) Histogram of buildings by block complexity. (**b**) Distribution of proposed new road construction by block complexity. (**c**) Kernel density estimates for distributions of building-block areal ratios at each value of *k* observed (denoted "complexity" in the figure) (**d**) Example of a high-*k* block whose morphology suggests bisection before reblocking is attempted as an appropriate upgrade.

We also see that most of the additional access construction—at least following our minimal intervention suggestions—was required in these relatively low complexity street blocks. The number of blocks in these circumstances in each city were numerous and involved the construction of several additional hundreds of kilometers of accesses in total. These physical requirements can be translated into budgets, given knowledge of local costs.

Blocks with many buildings and high complexity—often the prototypes for the most problematic urban slums—were in fact very rare in these two cities. Almost all street blocks with high complexity had similar or lower building densities than most other simpler blocks.

For extremely high-complexity blocks (e.g., Figure 9d), the nature of the problem of accessibility was in fact quite different from other urban compact neighborhoods. These were often sparsely occupied, peri-urban regions that were large in spatial extent, so that they may have benefited from an access structure that bisected or trisected the existing block, before a detailed access network for each building was attempted.

Importantly, some blocks with a high *k*-index may have limited access intentionally to inner parcels and were not at all informal settlements. Figure 8 shows the city block containing the Parliament building and other civic institutions of Freetown. This kind of campus (like military bases, hospitals, universities, etc.) was designed to limit free public access to internal parcels and was clearly not a slum. This type of inaccessible street block could be easily distinguished by the size, shape and orientation of buildings relative to patterns in true informal settlements.

### 3.2. Worldwide Analysis of Block Complexity

We have applied the analysis detailed above to most nations in Africa, Asia and Latin America. The results of our analysis are publicly available online at MillionNeighborhoods.org, where users can view the block-level complexity calculations based on existing OSM data across the entire Global South along with the street networks and building footprints underlying the calculations. Finally, all code to replicate the analysis for any source of building footprints and road networks is available as a Python3 package hosted on our open source code repository.

A snapshot of the map's current level of completeness is show in Figure 10. The geolocated block-level complexity files are available for free public download. We use this platform to share results with a variety of audiences, including local officials, researchers, and NGOs. Urban area and region specific analyses similar to those in Figures 8b and 9b can then be produced for other geographies, though data quality in OSM still varies widely. For example, while there are standards for which tags are used (driving our choice of tags in Table 1) [40], our exploration of the data suggests there exists heterogeneity in the degree to which these standards are enforced throughout the database. The changing nature of the built environment also represents a data quality challenge: are the building footprints visible on OSM at a given time representative of the physical buildings, given new construction and destruction? We argue that a *k*-index calculation for non-trivial blocks is likely robust to the absence of some buildings, given the density of building footprints in such blocks (e.g., Figure 6). While OSM represents the largest publicly available source of building footprint data, the completeness varies greatly by region and the slum areas which we target in our analysis can be especially data scarce. However, the size of the OSM building footprint database is constantly expanding and cases where the *k*-index is unrealistically low by the standards of local conditions due to missing footprints are valuable as measures of data completeness. Our work helps to highlight areas where community cartographic efforts may be focused.

One potential way to address the noise of OSM data is to fold in either local knowledge or alternative sources of building footprints or road networks. While we rely on OSM data, our method is agnostic to the precise source of the data and can be readily transferred to new data. As high-resolution satellite imagery and deep learning techniques for extracting geometries from images continue to expand, high quality building footprints and roads will continue to become more commonplace. We hope to compare the analysis we have presented so far to an application of the same methods to building and road location data from other cartographic sources in the near future.

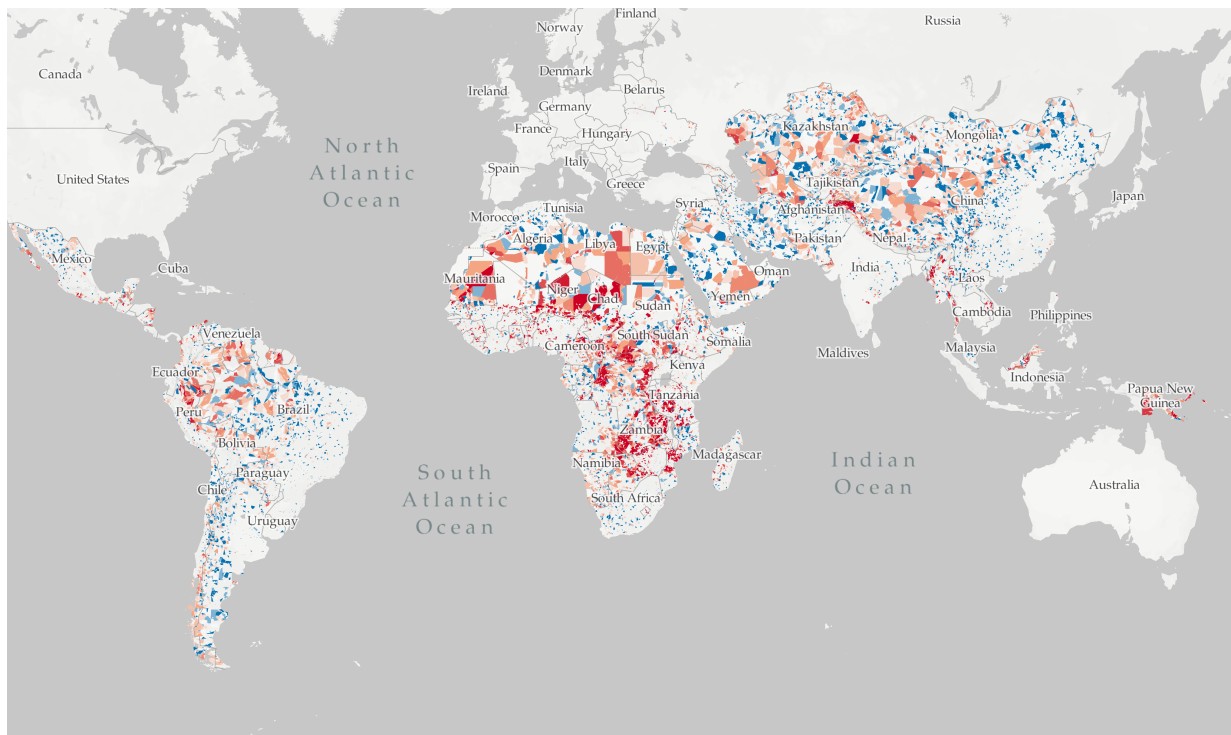

**Figure 10.** Snapshot of the MillionNeighborhoods.org project website, showing the street block level analysis for most nations in the global South. Countries with any colored polygons within their boundaries have been analyzed; at this zoom level, only the largest polygons are aggregated and shown. This image was exported from the Mapbox Studio backend using OpenStreetMap data, ©Mapbox, ©OpenStreetMap. To view the full interactive map, visit MillionNeighborhoods.org.

## 4. Discussion

We have shown how emerging high-resolution geospatial data on street networks and building footprints can be analyzed using topological methods to create a street block analysis of physical accessibility anywhere in the world. We have used a large dataset of crowdsourced data from OSM to show how these methods allow us to pinpoint infrastructure deficits in developing cities of the Global South, and estimate the complexity, cost and precise location of possible minimal solutions.

In particular, we motivated this work by the "challenge of slums" [3] in fast developing cities, especially in Africa and Asia. We recognize from the outset that the classification of a neighborhood as a "slum" often carries stigma [41] and is not binary. Nevertheless a necessary but not sufficient set of conditions for a neighborhood to be a slum deals with the lack of public street accesses, basic services, addresses and the availability of emergency services. For all these reasons—in resonance with processes of reblocking initiated by resident communities and local authorities [42,43]—we developed analytical and computational methods that are well tuned to different levels of difficulty in rendering an informal settlement universally accessible and thus help place it on the path of formal development with the rest of its city.

These methods allow us to analyze entire cities and regions block by block, breaking down what often appears as an insurmountable challenge into a set of simpler and better-defined local problems, many of them sharing common characteristics as we have seen for Freetown and Monrovia. While this approach is not meant to be a substitute to traditional urban planning, it provides residents, planners and policy makers with new insights and initial proposals that promote gradual upgrading. Thus, these methods are

best used as part of a systemic strategy that, while providing a uniform, universal level of access to anyone in the city, also respects and works along with local knowledge and history. These ingredients make for better cities in the long run, which preserve a human scale and historic character, as widely recognized by urbanists and historians [44,45].

In such processes, crowdsourced mapping becomes especially important and gains a new set of purposes and motivations. Most of the maps now available in OSM have been created only in the last few years, via a mixture of humanitarian work on the ground and around the world supported by new technologies, such as high-resolution remote sensing, GIS enabled mapping software and more recently machine learning [18]. While each of these methods contributes to the creation of better maps and associated information, we are excited about the prospect of retaining more and more of the local living experience, aspirations and agency in a collaborative global platform that can enable new spatial knowledge, civic values, modern urban planning and sustainable development anywhere in the world.

**Author Contributions:** Conceptualization, Anni Beukes and Luís M. A. Bettencourt; methodology, Satej Soman, Cooper Nederhood, Nicholas Marchio, and Luís M. A. Bettencourt; software, Satej Soman, Cooper Nederhood, and Nicholas Marchio; validation, Satej Soman, Cooper Nederhood, and Nicholas Marchio; formal analysis, Satej Soman, Cooper Nederhood, Nicholas Marchio and Luís M. A. Bettencourt; investigation, Satej Soman, Cooper Nederhood, Nicholas Marchio and Luís M. A. Bettencourt; resources, Satej Soman, Cooper Nederhood, Nicholas Marchio and Luís M. A. Bettencourt, data curation, Satej Soman, Anni Beukes, Cooper Nederhood, Nicholas Marchio, and Luís M. A. Bettencourt; writing—original draft preparation, Satej Soman, Anni Beukes, Cooper Nederhood, Nicholas Marchio, and Luís M. A. Bettencourt; writing—review and editing, Satej Soman, Anni Beukes, Cooper Nederhood, Nicholas Marchio, Luís M.A. Bettencourt; visualization, Satej Soman; supervision, Anni Beukes, Nicholas Marchio, and Luís M.A. Bettencourt; project administration, Anni Beukes, Nicholas Marchio, and Luís M. A. Bettencourt; funding acquisition, Anni Beukes, Nicholas Marchio, and Luís M. A. Bettencourt. All authors have read and agreed to the published version of the manuscript.

**Funding:** This research received no external funding.

**Acknowledgments:** We thank Christa Brelsford, Jamie Saxon, Annie Yang, Chris Barrington-Leigh, Devin White, Julia Koschinsky, Royal Mabakeng, and Smurti Jukur for discussions. We thank Yuxing Peng, Parmanand Sinha, Kazutaka Takahashi, and H. Birali Runesha of the University of Chicago Research Computing Center (RCC) for technical assistance in parallelization and computation. Prints use map data from Mapbox and OpenStreetMap and their data sources. To learn more, visit https://www.mapbox.com/about/maps/ and http://www.openstreetmap.org/copyright.

**Conflicts of Interest:** The authors declare no conflict of interest. The funders had no role in the design of the study; in the collection, analyses, or interpretation of data; in the writing of the manuscript, or in the decision to publish the results.

## Abbreviations

The following abbreviations are used in this manuscript:

| | |
|---|---|
| OSM | OpenStreetMap |
| LMIC | Low- and Middle-Income Country |
| SSA | Sub-Saharan Africa |
| GADM | Database of Global Administrative Areas |
| EO | Earth observation |
| OOA | object-oriented analysis |
| CNN | convolutional neural network |
| NGO | Non-Governmental Organization |
| GIS | Geographic Information System |

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
