# Peer review of "Worldwide Detection of Informal Settlements via Topological Analysis of Crowdsourced Digital Maps"

_ijgi, doi:10.3390/ijgi9110685_

Round 1

Reviewer 1 Report

Summary

The current work demonstrates a scalable and computational efficient method of using topological properties of digital maps to identify structural accessibility issues in cities. The method involves the integration of both geospatial analysis for initial data preparation and the incorporation of graph analysis for both determining current accessibility levels, and proposed low cost solutions for assisting city planners. The approach is simple as it is innovative and offers new insights into understanding structural imbalances in cities, particularly with a focus on those in the Global South. The paper was well written and is immediately understandable. I thank the authors for both tackling a problem that has a global footprint and can be understood at a trivial level for those that really need this type of analysis.

A few minor comments that I have no doubt that the authors can address within a short time follow. These are really directed at the data used, which while not the absolute focus of this effort compared to the development of the methodology, I still think at least some mention in the discussion is important.

Introduction

  • Line 22-23:
    • Reference 2 is far too outdated, please use one more updated.

Material and Methods

  • Table 1
    • There is a lot of heterogeneity in OSM tags, how do you  account for this when selecting road networks from the OSM data? I understand that in the cities that were demonstrated there has been a lot of work there, which I assume means validation of data. This could be briefly mentioned in the Discussion.
    • Similar question for building footprints.
  • Missing in-text reference to figures 2 and 3
  • Incompleteness in OSM is common problem especially when dealing with mobility corridors. How does this affect the use of graphs to both determine k-index complexity and to also come to a suggested optimal solution for spatial accessibility. In one scenario you can have the same road (or linestring) with a break that makes them disconnected. In a second scenario, could you not have problems of some areas not having non-conflated roads? Are these issues nullified or at low risk in the data that you use? If not, how are these accounted for in the network building and decomposition part of your methodology? I think some mention of this is at least required in the discussion.
  • Limitations of data missing as the previous comments suggest, especially as it relates to planetary scale deployment. You do mention “Worldwide” in the paper’s title so a discussion of the limits of the data is important.
  • Any plans for making a tools available to help planners working on smaller subsets of data? Or to make code accessible. I think makes the contribution even more high impact.

Discussion

  • How does you work complement other existing open data on cities? See some references in the following works:
    • Chakraborty, A., Wilson, B., Sarraf, S. and Jana, A., 2015. Open data for informal settlements: Toward a user׳ s guide for urban managers and planners. Journal of Urban Management4(2), pp.74-91.
    • Mahabir, Ron, Peggy Agouris, Anthony Stefanidis, Arie Croitoru, and Andrew T. Crooks. "Detecting and mapping slums using open data: a case study in Kenya." International Journal of Digital Earth13, no. 6 (2020): 683-707.

References:

UN, D. World urbanization prospects: The 2014 revision. United Nations Department of Economics and Social Affairs, Population Division: New York, NY, USA 2015, 41.

Author Response

Dear Reviewer,

Thank you for extensive review of our work. We are grateful for your insights.

We have made the following changes and corrections:

  • Updated reference 2
  • Added in-text references to figures 2 and 3 
  • Added discussion about heterogeneity of tags to section 3.2
  • Added a discussion of incompleteness of data to section 3.2
  • Added links to our open-source software repository
  • Folded the Mahabir paper into our discussion of prior work. The Chakraborty paper is interesting but we do not think there is a specific connection in a methods paper like this one to the guidelines for open data discussed. A discussion of how reblocking is being used on the ground would certainly intersect with the open data guide. 

Again, we appreciate your efforts in reviewing our work!

Reviewer 2 Report

Dear Authors,

Your paper on “Worldwide Detection of Informal Settlements via Topological Analysis of crowdsourced Digital Maps” provides a simple and convincing approach and database that provide physical proxies for the detection of informal areas using open and free datasets. The paper reads well and is a great contribution to move forward towards the provision of data on neglected urban spaces. I would have the following suggestion and requests to improve the paper before publication:

  • Section 2. Prior Work and Novel Contributions: a large part (of the first section) builds on bit dated literature, which should be mentioned before introducing the literature. Only late in this section, you cover the more recent literature on deep-learning and the departure from hand-crafted features etc.
  • Figure captions (e.g., Figure 2), I would suggest being more concise on what figures show and aiming for being less narrative in figure captions. Also, please add legends to the figures.
  • Line 122 and also later as part of your methodology: please be more explicit about the limitations of building data, which are often not well covered in OSM data, in particular in the Global South and often totally missing in informal areas. The examples you show in the paper, e.g., Kibera, are examples of exceptional good data coverage, because of large mapping projects.
  • The methodology section, I am missing a short section on how the validation was done. Could you please add information as part of the methodology and also results.
  • Linked to the previous comment, please also add more specific reflections on data limitations and what would be potential solutions to improve data on informal areas.
  • Figure 8 c, I do not fully understand what the number 0-17 represent? Could you please add information to the figure caption?
  • I would suggest to add a short conclusion paragraph, explaining the main contribution of your work in relation to filling global data gaps on informal urban areas.

Author Response

Dear Reviewer,

Thank you for the detailed  comments. We are grateful for your insights.

We have made the following minor changes:

  • Reduced the verbosity of figure captions. We believe with the more concise captions and in-text references, the figures are able to stand independently without the use of imprecise legends for illustrative and geometric data.
  • Addition of clarity to the caption of Figure 8 to explain that “complexity” refers to the k-index

The larger changes we have made include: 

  • Rewritten the section on prior work to focus on more recent explorations in deep learning
  • Added a note about validation in Section 3.2.  A larger and more systematic validation effort is underway and we will publish the results of that analysis when we conclude it. 
  • Expanded the section “Worldwide Analysis of Block Complexity” to include more discussion about the state of building footprint data in OSM and more broadly. Because we were already reflecting on data in this section, we thought this made most sense here

Regarding your suggestion about a short conclusion paragraph, we deem the contents of section 4 to cover the contribution of our work in relation to filling global data gaps on informal urban areas.

Again, we appreciate your efforts in reviewing our work!